# Efficient Synthesis of PVDF/PI Side-by-Side Bicomponent Nanofiber Membrane with Enhanced Mechanical Strength and Good Thermal Stability

**DOI:** 10.3390/nano9010039

**Published:** 2018-12-29

**Authors:** Ming Cai, Hongwei He, Xiao Zhang, Xu Yan, Jianxin Li, Fuxing Chen, Ding Yuan, Xin Ning

**Affiliations:** Industrial Research Institute of Nonwovens & Technical Textiles, College of Textiles & Clothing, Qingdao University, Qingdao 266071, China; 2016020202@qdu.edu.cn (M.C.); hehongwei@qdu.edu.cn (H.H.); 2017021351@qdu.edu.cn (X.Z.); qdyanx@qdu.edu.cn (X.Y.); 2016203960@qdu.edu.cn (J.L.); fxchen@qdu.edu.cn (F.C.)

**Keywords:** bicomponent electrospinning, side-by-side, PVDF/PI, nanofiber

## Abstract

Bicomponent composite fibers, due to their unique versatility, have attracted great attention in many fields, such as filtration, energy, and bioengineering. Herein, we efficiently fabricated polyvinylidene fluoride/polyimide (PVDF/PI) side-by-side bicomponent nanofibers based on electrospinning, which resulted in the synergism between PVDF and PI, and eventually obtained the effect of 1 + 1 > 2. Scanning electron microscopy (SEM) and Fourier transform infrared spectroscopy (FTIR) were used to characterize the morphology and chemical structure of nanofibers, indicating that a large number of side-by-side nanofibers were successfully prepared. Further, the thermal stability, mechanical strength, and filtration properties of PVDF/PI were carefully investigated. The results revealed that the bicomponent nanofibers possessed both good mechanical strength and remarkable thermal stability. Moreover, the mechanical properties of PVDF/ PI were strengthened by more than twice after the heat treatment (7.28 MPa at 25 °C, 15.49 MPa at 230 °C). Simultaneously, after the heat treatment at 230 °C for 30 min, the filtration efficiency of PVDF/PI membrane was maintained at about 95.45 ± 1.09%, and the pressure drop was relatively low. Therefore, the prepared PVDF/PI side-by-side bicomponent nanofibers have a favorable prospect of application in the field of medium- and high-temperature filtration, which further expands the application range of electrospun fiber membranes.

## 1. Introduction

Composite materials, due to their versatility over traditional single-component materials, have attracted great attention [1,2,3,4]. Composite fibers, such as islands-in-sea, sheath-core, and side-by-side, can be prepared using melt-blown, spunbonded, and electrospinning techniques [5,6,7,8], which are widely used in many fields, such as filtration, catalysis, energy, electronics, bioengineering, and so on [9,10,11,12]. Electrospinning is one of the most effective methods to obtain continuous composite nanofibers, which can efficiently and conveniently fabricate composite nanofibers with different structures [13,14,15,16]. The side-by-side bicomponent composite fiber, with two different polymers orienting parallel in one fiber, can be made by side-by-side electrospinning [17,18]. Compared to coaxial electrospinning, the two polymer fibers from the side-by-side electrospinning can be directly in contact with the environment, making it easier to embody and maintain the unique performance advantages of each component. In addition, the synergistic effect can be obtained by the complementarity and the correlation of the two components, and finally a 1 + 1 > 2 effect on the overall performance of polyvinylidene fluoride/polyimide (PVDF/PI) can be achieved [19,20,21].

In fact, the study of side-by-side electrospinning started relatively late, and there were not many reports about it until 2003, when Gupta and Wilkes [22] creatively prepared polyvinyl chloride/polyvinylidene fluoride (PVC/PVDF) and polyvinyl chloride/segmented polyurethane (PVC/Estane) bicomponent fibers for the first time using a parallel spinneret, which solved the problem of difficulty in blending two solutions to obtain a bicomponent fiber through co-electrospinning. From then on, research on the side-by-side electrospinning technology had been carried out mainly from the following two aspects [9,23,24]. On one hand, some functional nanocomposites with excellent performance were prepared. For example, TiO_2_/SnO_2_ bicomponent nanofibers fabricated by Liu’s group [25], employing parallel electrospinning, can simultaneously maximize the exposure of the two nanoparticles to the surface, thereby enhancing the quantum effect and improving the photocatalytic activity. Lv et al. [26] prepared C8-modified CeO_2_//SiO_2_ Janus fibers, which can make full use of the strong hydrophobicity of SiO_2_-C8 fiber and the affinity of CeO_2_ to phosphate group, achieving favorable selective biological separation. On the other hand, researchers carefully studied the effects of experimental conditions on the side-by-side fiber structure. For instance, polyacrylonitrile/polyurethane (PAN/PU) bicomponent fibers with high crimp were fabricated by Lin et al. [27] using polymers with different retractilities. In addition, obtaining a high proportion of parallel fibers has been a difficulty in spinning. Zhu’s team [28] discussed in detail the effects of five different needles on parallel spinning. It was found that the effective contact between the two fluids at the tip of the needle to be a very important condition for obtaining a high proportion of side-by-side fibers. Yu’s group [29] designed a new type of spinneret, comprising two acentric needles nested into a third metal capillary, for the development of a three-fluid electrospinning process. The high-quality polyvinylpyrrolidone/shellac Janus nanofibers can be gained by the external solvent surrounding two core fluids arranged side-by-side. In 2018, Kan’s group [30] used the finite element method (FEM) to simulate the electric field distribution of side-by-side electrospinning, and studied the influence of spinneret wall thickness, spinning voltage, and receiving distance on the distribution of the electrostatic field.

In this work, the spinneret of side-by-side electrospinning is ameliorated to counteract the electrostatic repulsion of the two jets by applying the same electric field to the outside of the parallel needles. This improvement can ensure a close contact between the two polymer solutions, so that a large number of PVDF/PI side-by-side bicomponent fibers can be fabricated efficiently. PVDF, as a high-tensile-strength product in fluorine-containing resins, is easy to form a membrane during electrospinning. Simultaneously, PVDF possesses strong hydrophobicity and UV radiation resistance, and it is often used to prepare filter membranes [31]. However, the melting point of PVDF is generally between 140 and 170 °C, which limits the application of the traditional PVDF electrospun membrane in the field of high-temperature filtration, catalysis, and energy [32,33]. Polyimide has excellent thermal stability, and its high-heat resistance reaches above 400 °C. Recently, PI nanofibers prepared by electrospinning have been used in microelectronics, lithium ion batteries, and in other fields [34,35]. Based on the above information, the PVDF/PI bicomponent fiber obtained in this study combines the performance superiority of these two polymer materials. The fabrication method employed in our work is simple and efficient, and the performance of the composite fiber material is excellent. Interestingly, the mechanical properties of the PVDF/PI fiber membranes effectively improved after the heat treatment. At 230 °C, the mechanical strength of the PVDF/PI fiber membrane increased by 2.13 times (7.28 MPa at 25 °C, 15.49 MPa at 230 °C). This is mainly due to the melting of PVDF in the side-by-side bicomponent fibers, resulting in the formation of bonding points and strengthening the associations between the fibers. With PI acting as a supporting skeleton, the PVDF/PI fiber maintained a good fiber morphology during the high-temperature processing. Therefore, PVDF/PI membranes can still exhibit a good fiber micro-morphology, porous structure, and filtration performance after treatment at 230 °C, which is expected to be applied in the field of high-temperature filtration. Furthermore, the PVDF/PI side-by-side bicomponent fiber membrane displays good mechanical properties and excellent thermal-dimensional stability, which also has certain application prospects in the field of lithium batteries.

## 2. Materials and Methods

### 2.1. Materials

Polyvinylidene fluoride hexafluoropropylene (PVDF-HFP, M_w_ = 400,000) was purchased from Aladdin Reagent Co., Ltd. (Shanghai, China). Polyimide (PI) resin was supplied by the Hangzhou Plastic UNITA Special Technology Co., Ltd. (Hangzhou, China). Dimethylformamide (DMF) and acetone were obtained from the Sinopharm Chemical Reagent Co., Ltd. (Shanghai, China). All chemical reagents were used as received without further purification.

### 2.2. Side-by-Side Electrospinning

The PVDF electrospinning solution (16 wt %) was prepared by dissolving the PVDF particles in the mixed solvent of DMF/acetone (1/1 in weight), magnetic stirring for 24 h at room temperature. Additionally, 15 wt % PI solution in DMF was prepared for electrospinning. Figure 1 depicts the schematic of our homemade side-by-side electrospinning device, including a high-voltage DC supply (DW-P503-1ACF0, Dongwen High Voltage Power Supply (Tianjin) Co., Ltd., Tianjin, China), a dual-channel propulsion pump (LSP02-1B, Baoding Lange Constant Flow Pump Co., Ltd., Baoding, China), and a homemade roller receiver. As observed in Figure 1, the PVDF and PI electrospinning solutions were separately added to two syringes, which formed two mutually contacting Taylor cones on the parallel needle under the action of electrostatic field. When electrospinning was used to fabricate composite fibers with side-by-side structures, the key was that the two polymer solutions were ejected at the same rate, and the effective contact between two Taylor cones were formed at the outlet of the dual-spinneret. Here, in order to offset the electrostatic repulsion of the two jets, we applied the same electric field to the outside of the parallel needle, as described in detail in Figure 1. In this case, it was possible to ensure that the ejected droplets at the tip of dual-spinneret were in sufficient contact. As a result, the separation tendency of the two jets in the spinning process was greatly reduced, and most of the obtained fibers had the side-by-side morphology (as detailed in the Appendix A).

Many process parameters for electrospinning, such as bias voltage, working distance, and feed rate, had an absolute effect on the fiber formation and the fiber quality. After optimization, the voltage was fixed at 18 kV, the distance between tip and collector at 17 cm, and the flow rates of both the PI and PVDF solutions were set to 0.5 mL/h. The whole electrospinning experiment was carried out under ambient conditions (24 ± 2 °C, 48 ± 6% RH). The obtained electrospun fiber membranes were placed in a vacuum oven at 60 °C for 1 h to remove the solvent remaining on the surface. In order to further test the filtration efficiency of the membranes before and after the high-temperature treatment, a stainless-steel mesh (25 cm × 25 cm, a pore size of about 1 mm) was selected as a receiver to prepare the filter material.

### 2.3. Measurement and Characterizations

Scanning electron microscopy (SEM, Phenom Pro SEM, Hitzacker, Germany) was employed to investigate the morphology and structure of the fiber products. The samples were subjected to gold sputter-coating in a vacuum prior to imaging (SBC-12, Beijing KYKY Technology Co., Ltd., Beijing, China). The average sizes (diameters of monolithic nanofibers) were determined by measuring the fibers at more than 100 different places in the SEM images, using Adobe Acrobat 9 Pro software (San Jose, CA, USA). The nanofiber structure was also characterized by Fourier transform infrared spectroscopy (Nicolet iS5, Thermo Scientific, Waltham, MA, USA) in attenuated total reflection (ATR) mode, and the wavenumber ranged from 500 to 2500 cm^−1^.

The mechanical property of nanofiber membrane was investigated using the universal tensile testing machine (Instron 6025, Jinan Liangong Testing Technology Co., Ltd., Jinan, China), the sample size was 8 mm × 60 mm, the distance between the two clamps was 20 mm, and the stretching rate was 5 mm/min. Each group of 5 samples was tested and averaged.

The filtration measurements for the selected membranes were performed using an air filtration device (TOPAS AFC-131, Frankfurt, Germany). In the filtration test, the electrospun nanofiber membrane was cut into a circular sample with a diameter of 17 cm, and the filtration pressure drop and filtration efficiency of aerosol particles with 0.3 μm diameter were obtained under the air-flow rate of 32 L/min. The overall filtration performance of the fiber membrane was evaluated by the calculated quality factor (QF) value [14]. The maximum and average pore sizes were determined according to the bubble method using aperture tester (TOPAS PSM-165, Frankfurt, Germany) with a fixed testing area of 2.01 cm^2^. All data were tested in at least 5 sets of samples and averaged.

## 3. Results and Discussion

### 3.1. Morphology and Chemical Structure

The key to obtain side-by-side fibers in electrospinning was to ensure that the two fluid jets with different properties were not separated in the electric field [29,36]. In the experiment, two kinds of solvent with a good compatibility were specially selected to configure electrospun solution to avoid the difficulty in contact during confluence. More importantly, since both polymer solutions carried the same electrostatic charge, it was easy to cause repulsion and lead to fiber separation. Therefore, electrospinning had high requirements for spinneret, voltage, and the speed of each fluid. A large number of experimental results showed that the fiber diameter distribution was uniform and the side-by-side results were the best when the propelling speed of both solutions was 0.5 mL/h, the experimental voltage was 18 kV, the receiving distance was 17 cm, and the improved spinneret was used simultaneously. The scanning electron microscopy (SEM) image (Figure 2c,f) shows the formation of PVDF/PI side-by-side bicomponent fibers with an average diameter of 541 ± 40 nm. It can be clearly seen in the enlarged view that the PVDF/PI are tightly bonded by two single-component fibers with grooves at the interface of the two-phase fibers. To further verify whether the fibers in the seemingly side-by-side state were bicomponent fibers, a fluorescent labelling method was intentionally adopted [36], as detailed in the Appendix A. It was observed from the fluorescence microscopy image (Appendix A) that the side-by-side fibers were formed, and were composed of PVDF and PI. In addition, the single-component electrospun PVDF fiber and PI fiber had a smooth surface (Figure 2a,b). As shown in the Figure 2d–e, compared with PI fibers (average diameter 467 ± 50 nm), PVDF fibers have a smaller average diameter (223 ± 20 nm). Simultaneously, the PVDF fibers were stacked more tightly than PI, as shown in Figure 2a.

The FTIR was used for confirming the successful synthesis of PVDF, PI, and PVDF/PI nanofibers, which characterized the chemical structure of the respective fiber membranes. In the spectrum of pure PVDF fiber, the absorption peak at 1174 cm^−^^1^ corresponds to the CF_3_ vibration. The absorption peak at 1400 cm^−^^1^ corresponds to the deformation vibration of the CH_2_ group. Moreover, the peaks around 840 cm^−^^1^ is related to the vibration of the PVDF crystal phase [37], as shown in Figure 3a. It can be seen from Figure 3b that the characteristic absorption peak representing the chemical structure of the PI molecule, which is the absorption peak of the imide ring, such as 1770, 1720, and 720 cm^−^^1^, appears in the spectrum of pure PI fiber. The absorption peak at 1360 cm^−^^1^ corresponds to the C–N stretching in the imide ring [1,38]. After combining the PI and the PVDF by side-by-side electrospinning, the combined characteristic peaks of PI and PVDF can be found in the FTIR of bicomponent fiber membrane (Figure 3c). It is demonstrated that there is no chemical structural change in the two polymers during electrospinning, and the bicomponent fibers are composed of PVDF and PI.

### 3.2. Thermal Stability

Next, the thermal stability of PVDF, PI, and PVDF/PI electrospun membranes were carefully studied (Figure 4). Therefore, it can be found that before and after the heat treatment, there is no obvious change in the PI fiber membrane, and the original fiber appearance and structure are still maintained, as shown in Figure 4a,b,e,f. It also proves that the PI membrane obtained by electrospinning has a good thermal stability. In contrast, the PVDF membrane shrinks rapidly and the area decreases significantly when the temperature rises to 140 °C (Figure 4a,b). Moreover, when the temperature reaches 230 °C, the PVDF completely melts into a transparent membrane. It can be seen from SEM (Figure 4c) that the PVDF membrane expands and melts at 140 °C, and some pores are blocked up by the melted PVDF. After heat treatment at 230 °C, the PVDF has completely lost the fiber configuration (Figure 4d). Furthermore, the PVDF/PI fibers have a good thermal-dimensional stability, and the changes of the PVDF/PI membrane area are negligible, as displayed in Figure 4b. This is mainly attributed to the fact that the high-temperature-resistant PI maintains the complete skeleton of the fiber. Moreover, it can be observed via SEM (Figure 4g,h) that the PVDF in PVDF/PI is thermally fused after the heat treatment, and the fibers are bonded to each other to form a bonding point (such as A and B). We speculate that this change will strengthen the mechanical properties of the composite fiber materials. Therefore, the mechanical strength, filtration performance, and pore size of the three samples at the normal temperature of 25 °C and after heat treatment at 140 and 230 °C were analyzed.

### 3.3. Thermo-Mechanical Properties

Stress–strain curves of PVDF, PI, and PVDF/PI electrospun membranes before and after the heat treatment were measured to evaluate the mechanical property of the samples, as shown in Figure 5. At the normal temperature of 25 °C, the PVDF membrane had a good ductility, and its elongation at break could reach 142.3%. At the same time, the pile up between the fibers was tight (Figure 2a), resulting in the tensile strength of 7.98 MPa. After heating at 140 °C, the PVDF was thermally melted, and the fibers were bonded to each other (Figure 4c), and the strength increased to 15.70 MPa. When heated to 230 °C, the PVDF melted completely, and its strength was about 26.04 MPa. However, the PI fiber membrane obtained by electrospinning was loose in structure and easy to disassemble. Thence, the elongation at break and tensile strength of PI membrane obtained by electrospinning were only 33.65% and 3.16 MPa, respectively. As the heat-treatment temperature increased, the molecular relaxation was induced, resulting in a slight decrease in strength [39]. The strength of the membrane was 2.91 MPa at 140 °C. When the temperature reached 230 °C, the strength was reduced to 2.63 MPa. In contrast, the PVDF/PI bicomponent fiber membrane obtained by the side-by-side electrospinning has the advantages of two single-component fiber membranes, exhibiting excellent mechanical properties and a good thermal stability. Through the heat treatment, the breaking strength of PVDF/PI fiber membrane increased more than twice, from 7.28 MPa at the beginning to 12.36 MPa (140 °C) and 15.49 MPa (230 °C). This result is mainly attributed to the melting of the PVDF fibers by heating and the formation of bonding points between the fibers (Figure 5d), which strengthens the bond between the fibers. However, slippage is easily generated between PVDF/PI fibers before the heat treatment [40]. Fortunately, the PVDF/PI membrane can maintain a good pore structure, creating conditions for its application in the field of high-temperature filtration.

### 3.4. Filtration and Pore Size

A good thermal stability was also an important property for filter materials, especially for those used in high-temperature environment. The filtration efficiency and permeability of PVDF, PI, and PVDF/PI samples treated with 30 min at different temperatures were studied, and the results are given in Figure 6. As can be seen, the PVDF (with a basis weight of 2.73 g/m^2^) emerged with a high-filtration efficiency and a pressure drop at room temperature (Figure 6a,b). However, with the increase in temperature, the filtration efficiency of PVDF decreased significantly, and the filtration capacity was completely lost at 230 °C. This is mainly due to the fact that the thermal deformation temperature of PVDF is about 100 °C, and its melting point is 167 °C. As a result, the fibers begin to melt and the filtration efficiency drops sharply as the temperature increases. Furthermore, the PI (with a basis weight of 3.07 g/m^2^) and the PVDF/PI (with a basis weight of 3.32 g/m^2^) membranes showed similar filtration efficiency at room temperature. The filtration efficiency of the PI fiber membrane decreased slightly, and it was 91.93 ± 1.35% at 230 °C. Interestingly, the PVDF/PI bicomponent fiber membrane still maintained a high-filtration efficiency after the heat treatment, with a value of 95.45 ± 1.09% at 230 °C. The pressure drop of PI and PVDF/PI membranes corresponded to the change in the filtration efficiency, which is also in agreement with the change in the pore size of the fiber membranes as shown in Table 1. The pore size of the filter material largely determines the performance of the membrane. From the detailed porous structures of the fiber membranes (Table 1), we conjecture that the increase in the filtration efficiency and the pressure drop of the PVDF/PI membrane was mainly due to the reduction in the pore size caused by the partial PVDF swelling, which enhanced the screen resistance to particulate matter. The quality factor (QF) value is the performance index to evaluate the filtration efficiency and pressure drop synthetically. As shown in Figure 6c, the PVDF/PI membrane clearly exhibits a higher QF value than the single-spinning at high temperature, showing an excellent filtration effect. The air permeability of the filter material affects the filtration pressure drop and service life to some extent. At high temperatures, the PVDF/PI membrane still maintains good air permeability (Figure 6d). Based on the above analysis, compared with pure PVDF and PI electrospun fiber membranes, the PVDF/PI side-by-side bicomponent fiber membranes are more suitable for high-temperature filtration.

## 4. Conclusions

In summary, we efficiently fabricated the PVDF/PI bicomponent fibers using a side-by-side electrospinning technology. The obtained bicomponent fiber membrane possesses the high strength of PVDF and the heat resistance of PI, and finally exhibits attractive mechanical properties and a good thermal stability. In the PVDF/PI fibers, the PVDF with a low melting point melts during the heat treatment, resulting in the formation of adhesion between the fibers, and subsequently increasing the strength of the fiber membrane by more than twice. More importantly, PVDF/PI membrane can still maintain fine pore structure after the heat treatment. Through the analysis of filtration performance, it is shown that the PVDF/PI side-by-side bicomponent fiber membrane can maintain excellent filtration performance and good mechanical properties after high-temperature treatment, and is expected to be applied in the field of medium- and high-temperature filtration.

## Figures and Tables

**Figure 1 nanomaterials-09-00039-f001:**
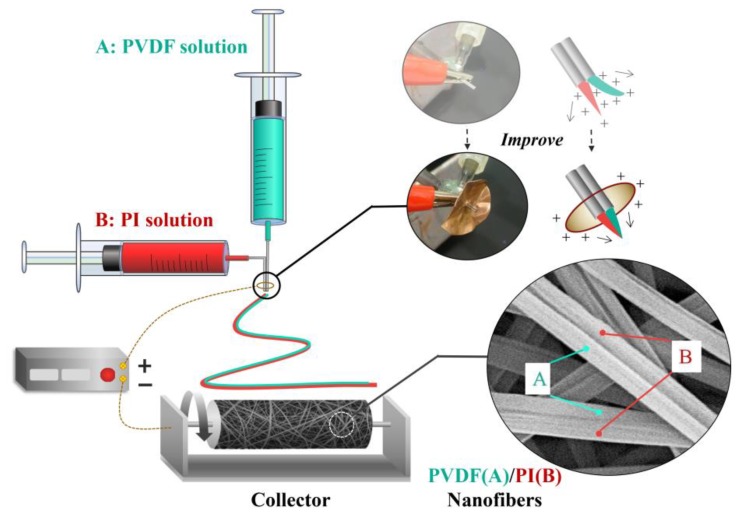
Schematic diagram of side-by-side bicomponent electrospinning equipment.

**Figure 2 nanomaterials-09-00039-f002:**
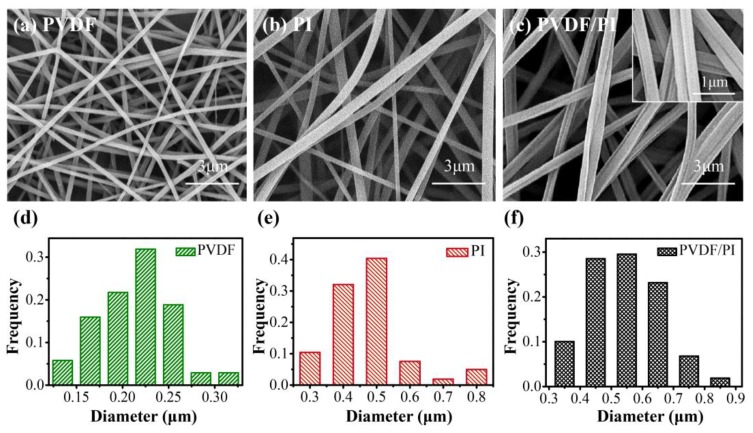
SEM images of PVDF (**a**), PI (**b**), and PVDF/PI (**c**) nanofibers; Fibers diameter distribution images of PVDF (**d**), PI (**e**) and PVDF/PI (**f**).

**Figure 3 nanomaterials-09-00039-f003:**
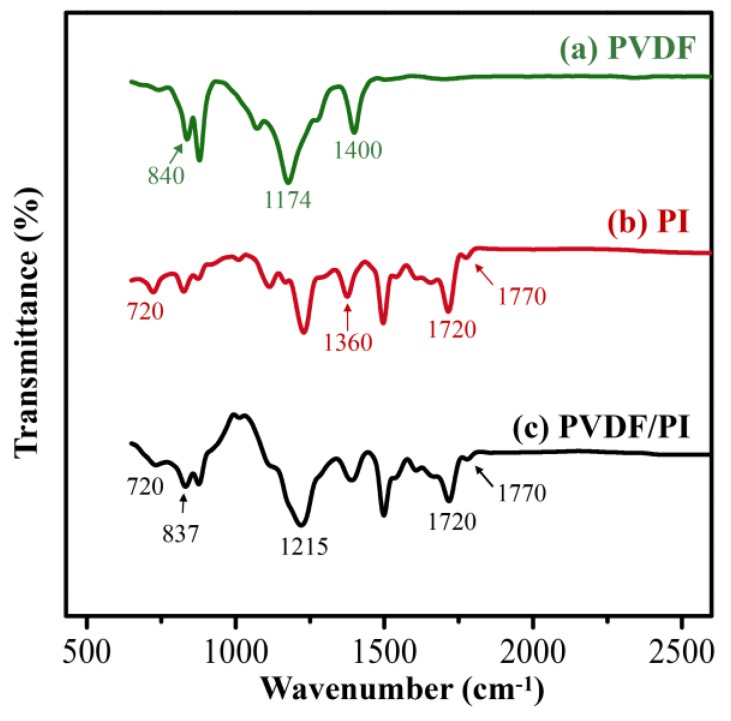
The FTIR spectra of PVDF (**a**), PI (**b**), and PVDF/PI (**c**) nanofibers.

**Figure 4 nanomaterials-09-00039-f004:**
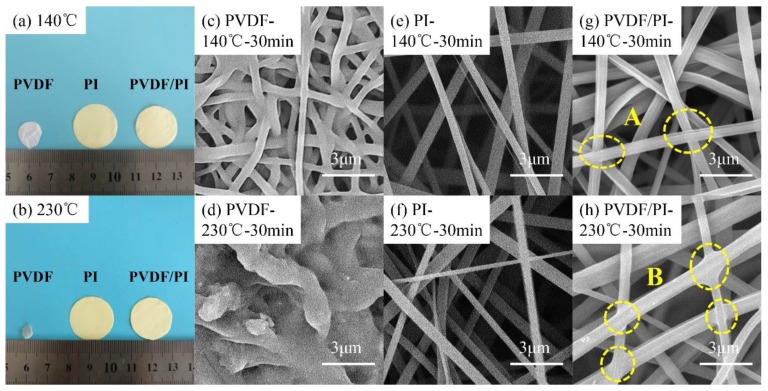
Photograph (**a**,**b**) and SEM images of PVDF (**c**,**d**), PI (**e**,**f**), and PVDF/PI (**g**,**h**) electrospun membranes before and after heat treatment.

**Figure 5 nanomaterials-09-00039-f005:**
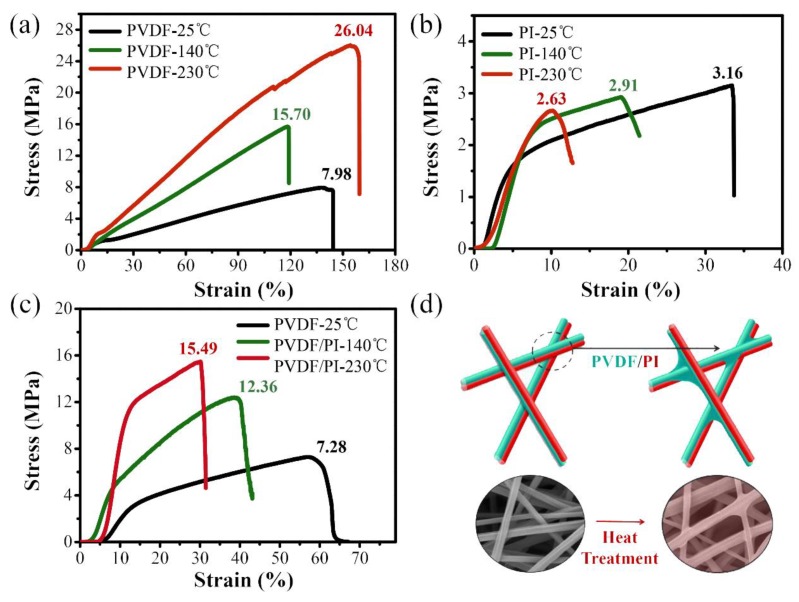
The stress–strain curves of PVDF (**a**), PI (**b**), and PVDF/PI (**c**) electrospun membranes before and after the heat treatment; Fibers heat treatment diagram (**d**).

**Figure 6 nanomaterials-09-00039-f006:**
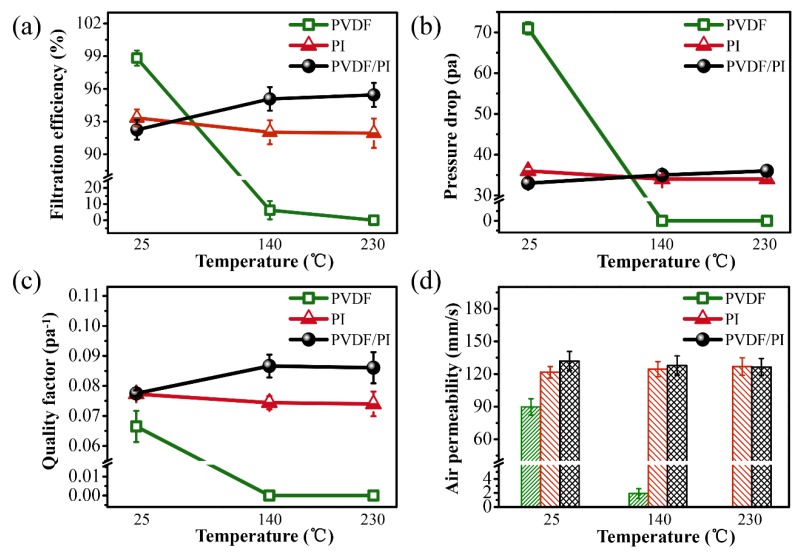
Filtration efficiency (**a**), pressure drop (**b**), quality factor (QF) (**c**), and air permeability (**d**) of PVDF, PI single-component fiber membrane and PVDF/PI bicomponent fiber membrane at 25 °C, 140 °C and 230 °C, respectively.

**Table 1 nanomaterials-09-00039-t001:** Maximum pore size and average pore diameter of PVDF, PI, and PVDF/PI at 25 °C, 140 °C and 230 °C, respectively.

Samples	Thickness (μm)	Maximum Pore Size (μm)			Average Pore Size (μm)		
		25 °C	140 °C	230 °C	25 °C	140 °C	230 °C
**PVDF**	29.7 ± 1.2	1.968	1.755	-	1.402	0.818	-
**PI**	30.5 ± 2.3	5.463	5.977	6.301	3.241	3.886	3.971
**PVDF/PI**	31.3 ± 1.9	5.662	5.463	5.324	3.587	3.171	2.903

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
