# Peer review of "Efficient Synthesis of PVDF/PI Side-by-Side Bicomponent Nanofiber Membrane with Enhanced Mechanical Strength and Good Thermal Stability"

_nanomaterials, 2018, doi:10.3390/nano9010039_

Reviewer 1 Report

The authors used two needles to create an electrospun mat, which they claim contains a "side-by-side" morphology. When performing the mechanical testing, it seems that the authors did not account for the void fraction in each mat, which will have a substantial influence on the stress values obtained for each mat (especially since the PVDF fibers are noticeably smaller than the other two systems). Samples were imaged with SEM and filtration performance was measured. The authors claim that via heat treating, the strength of the PVDF/PI fiber mat is doubled while observing no loss in filtration efficiency.

I am not convinced a large fraction of the "side-by-side" fibers actually possess a side-by-side morphology. The SEM image in Figure 1 uses false coloring so it is very difficult to tell if there is actually a side-by-side structure from the image. A number of the false colored images in Figure 1 also appear to be cylindrical with just one circular circumference.

Examining Figure 2, the morphology of the PVDF and PI fibers looks good, but then in the figure 2C I only see one fiber (directly in the middle moving down the figure vertically) that can be considered to have a side-by-side morphology. All of the other fibers in the image appear to be smooth and cylindrical again. In fact, figure S1 also shows only a few side-by-side fibers and even shows where some of those fibers were seperated (particularly at the bottom). The figure S2 uses flourescence imaging but doesn't contain a scale bar and only shows 3 fibers which appear to be welded together.

In line 182, it is claimed from the FT-IR that "the properties of the material were retained". I'm not sure how this claim is supported. Which properties? All the information given here is about the chemical structure of PVDF, PI, and the bicomponent fibers. Figure 3 also does nothing to confirm a true side-by-side morhpology. If both PVDF and PI were dissolved in DMF, it is possible that while spinning, the two jets mix to create a physical blend in a single circular fiber? And the few observed fibers that appear to be "side-by-side" are the exception? More microscopy over a larger areas and examining more fibers is needed before the "side-by-side" morphology is confirmed.

The mechanical interpretation needs re-done. The authors measure Force vs. Displacement on the Instron and then likely divide Force/Nominal cross section area to obtain Stress. The likely also divide Displacement/gauge length to obtain strain. They should also correct for the void fraction in the nominal cross-section area. Note that these are porous structures and only the physical occupied area should be used to obtain an appropriate value for stress. Since the authors did not do this, none of the results can truly be interpreted on a molecular level as the authors do. I also do not see error or standard deviation with these measurements so it is unclear how statistically significant any of the mechanical results are.

For pore size and filtration efficiency, are these measurements all based on 1 single production run? Or were multiple mats fabricated with multiple runs. This should be clarified, and these analyses should be performed on multiple different mats.

The authors also mention industrial applications being the interest of high temperature applications (lines 236-238), but choose a fabrication technique that manufactures small mats at a very low throughput. They should comment further on scale up ability of the side-by-side electrospinning process (which has many limitations) or remove this discussion. It is quite misleading about the impact of the work.

Author Response

Response to Reviewer 1 Comments

The authors used two needles to create an electrospun mat, which they claim contains a "side-by-side" morphology. When performing the mechanical testing, it seems that the authors did not account for the void fraction in each mat, which will have a substantial influence on the stress values obtained for each mat (especially since the PVDF fibers are noticeably smaller than the other two systems). Samples were imaged with SEM and filtration performance was measured. The authors claim that via heat treating, the strength of the PVDF/PI fiber mat is doubled while observing no loss in filtration efficiency.

Thanks for your professional comments on this manuscript, which are very instructive to improve our article and research. We have carefully studied your comments and made corresponding amendments. We hope to obtain your approval. Here is our reply:

Point 1: I am not convinced a large fraction of the "side-by-side" fibers actually possess a side-by-side morphology. The SEM image in Figure 1 uses false coloring so it is very difficult to tell if there is actually a side-by-side structure from the image. A number of the false colored images in Figure 1 also appear to be cylindrical with just one circular circumference.

Response 1: Thanks for the critical suggestion. In Figure 1, the artificially added color was originally intended to make the two-component distinction more visible and to help the readers better understand it. We are sorry for the bad result. In order to avoid disturbing the original appearance of the electron microscope and causing controversy, we removed the false color of Figure 1 in the manuscript and chose the more clearly recognizable SEM image as the representation image.

Point 2: Examining Figure 2, the morphology of the PVDF and PI fibers looks good, but then in the figure 2C I only see one fiber (directly in the middle moving down the figure vertically) that can be considered to have a side-by-side morphology. All of the other fibers in the image appear to be smooth and cylindrical again. In fact, figure S1 also shows only a few side-by-side fibers and even shows where some of those fibers were separated (particularly at the bottom). The figure S2 uses fluorescence imaging but doesn't contain a scale bar and only shows 3 fibers which appear to be welded together.

Response 2: In our experiments, DMF is used as a solvent to dissolve PVDF and PI resin. The co-solvent is favour for obtaining side-by-side bicomponent fibers (Polymer, 2003, 3, 6353-6359; Applied Surface Science 2014, 319, 21-28; Colloids and Surfaces B: Biointerfaces 2016, 138, 110-116). Meanwhile, it also causes a certain degree of fusion between the two components at the contact surface, resulting in the interface is not very clear. Finally, the side-by-side fibers are formed and component A and B not separated. We are sorry that in the SEM image, the side-by-side morphology is not particularly noticeable when viewing the fibers from the side. It can be seen from Figure 2 that the PVDF and PI fibers exhibit a very smooth cylindrical fiber surface without any grooves, while the PVDF/PI side-by-side bicomponent fibers clearly have grooves that were thought as two-phase interfaces. To further verify whether the fibers in the seemingly side-by-side state are bicomponent fibers, we have deliberately adopted a fluorescent labelling method, which different coloring fluorescent agents were added into the electrospinning precursor solutions of PVDF and PI, respectively (Biomed Microdevices,2014,16,793-804). It was observed from the fluorescence microscopy image (Figure S2.) that the side-by-side fibers were indeed formed, and composed of PVDF and PI. The scale bar problem in the fluorescence imaging mentioned by the reviewer is revised.

Point 3: In line 182, it is claimed from the FT-IR that "the properties of the material were retained". I'm not sure how this claim is supported. Which properties? All the information given here is about the chemical structure of PVDF, PI, and the bicomponent fibers. Figure 3 also does nothing to confirm a true side-by-side morphology. If both PVDF and PI were dissolved in DMF, it is possible that while spinning, the two jets mix to create a physical blend in a single circular fiber? And the few observed fibers that appear to be "side-by-side" are the exception? More microscopy over a larger areas and examining more fibers is needed before the "side-by-side" morphology is confirmed.

Response 3: We are very sorry that there was a problem with the description of FT-IR spectra in the article, which has been modified. Although both PVDF and PI solutions contain DMF, they cannot be completely mixed together because of the short contact time of the two jets during the whole spinning process. In addition, in the support materials, we have given a clearer equipment improvement program. As can be seen from Figure S1c, there are some single-component fibers that exhibit a smooth cylindrical appearance before the equipment was improved. Furthermore, the improvement of the equipment greatly reduces the separation tendency of the two components in the spinning process, and most of the obtained fibers have side-by-side morphology (Figure S1d). It is undeniable that there is still separation after the improvement of the equipment. In the future, we will further study how to increase the proportion of side-by-side bicomponent fibers and the influence factors on side-by-side morphology.

Point 4: The mechanical interpretation needs re-done. The authors measure Force vs. Displacement on the Instron and then likely divide Force/Nominal cross section area to obtain Stress. The likely also divide Displacement/gauge length to obtain strain. They should also correct for the void fraction in the nominal cross-section area. Note that these are porous structures and only the physical occupied area should be used to obtain an appropriate value for stress. Since the authors did not do this, none of the results can truly be interpreted on a molecular level as the authors do. I also do not see error or standard deviation with these measurements so it is unclear how statistically significant any of the mechanical results are.

Response 4: Thanks for the critical suggestion, and we have revised the description of the mechanical properties. In terms of mechanical strength, we have chosen the stress-strain test. These results are based on the experimental data of many samples and are illustrated in the experimental section. As the reviewer said, the stress is obtained by the Force/Nominal cross section area and the strain is obtained by the Displacement/gauge length (Journal of Colloid and Interface Science, 2014, 428, 18-26; Chemical Engineering Science, 2019, 193,230-242; New Journal of Chemistry, 2015, 39, 7797-7804; Polymer, 2004, 45, 1895-1902). We do agree to the reviewer’s opinion on the porosity effect when measuring the stress/strain of e-spinning film on the Instron. 5 groups of testing data for every sample have been adopted including control sample to reduce an error and make the data comparable, which coincided with those references mentioned above. In addition, we still revised the manuscript and try to make its description more rigorous.

Point 5: For pore size and filtration efficiency, are these measurements all based on 1 single production run? Or were multiple mats fabricated with multiple runs. This should be clarified, and these analyses should be performed on multiple different mats.

Response 5: We are so sorry for the lack of detail in the manuscript, which led to misunderstanding of pore size and filtration efficiency data. In fact, the pore size and filtration efficiency in the manuscript are obtained from repeated experiments of many samples, and the average values are given in this paper. In order to avoid misunderstanding, we have made an amendment based on the comments of reviewer, and added a specific description to the article.

Point 6: The authors also mention industrial applications being the interest of high temperature applications (lines 236-238), but choose a fabrication technique that manufactures small mats at a very low throughput. They should comment further on scale up ability of the side-by-side electrospinning process (which has many limitations) or remove this discussion. It is quite misleading about the impact of the work.

Response 6: Thanks for your criticism. Although the properties of PVDF/PI make it have a potential application in the field of high temperature filtration, there is still a long way to go in the field of industrial filtration. We apologize for the inaccuracy and sincerely accept the reviewer’s suggestion to revise the manuscript.

Reviewer 2 Report

This paper is potentially good. For the benefit of readers, some points need clarifying. These are given below.

1) The focus of this study would be to prepare side-by-side bicomponent nanofibers by applying adequate electric field. Adding micrographs of the dual-spinneret and the voltage-applying copper ring to Fig. 1 should help readers to grasp the details.

2) The method for IR spectra measurements should be described more determinably. Since the figure showing the IR spectra (Fig. 3) is drawn in low resolution, finer spectra should be presented. Assignments of the absorption bands should be based on sound literatures.

Author Response

This paper is potentially good. For the benefit of readers, some points need clarifying. These are given below.

Thanks for your professional review and recognition of the work. We have carefully studied your comments and made corresponding amendments. We hope to obtain your approval. Here is our reply:

Point 1: The focus of this study would be to prepare side-by-side bicomponent nanofibers by applying adequate electric field. Adding micrographs of the dual-spinneret and the voltage-applying copper ring to Fig. 1 should help readers to grasp the details.

Response 1: Thanks for your professional advice. We modified Figure 1 and added detailed pictures of the experimental equipment to the supporting material so that the reader could better grasp the details.

Point 2: The method for IR spectra measurements should be described more determinably. Since the figure showing the IR spectra (Fig. 3) is drawn in low resolution, finer spectra should be presented. Assignments of the absorption bands should be based on sound literatures.

Response 2: Thanks for the critical suggestion. We revised the manuscript to describe in detail the method for IR spectra measurements. Simultaneously, the samples were tested and plotted again to provide a more detailed IR spectrum. In addition, we have added references to identify the absorption peaks.

Round  2

Reviewer 1 Report

The authors have addressed all issues raised in my original review.

Reviewer 2 Report

The author has adequately edited the manuscript. It is acceptable now.